# Differentially CTCF-Binding Sites in Cattle Rumen Tissue during Weaning

**DOI:** 10.3390/ijms23169070

**Published:** 2022-08-13

**Authors:** Clarissa Boschiero, Yahui Gao, Ransom L. Baldwin, Li Ma, Cong-jun Li, George E. Liu

**Affiliations:** 1Animal Genomics and Improvement Laboratory, Beltsville Agricultural Research Center, Agricultural Research Service, U.S. Department of Agriculture, Beltsville, MD 20705, USA; 2Department of Animal and Avian Sciences, University of Maryland, College Park, MD 20742, USA

**Keywords:** cattle, ChIP-seq, CTCF, epithelial tissue, regulatory elements, rumen development, weaning

## Abstract

The weaning transition in calves is characterized by major structural changes such as an increase in the rumen capacity and surface area due to diet changes. Studies evaluating rumen development in calves are vital to identify genetic mechanisms affected by weaning. This study aimed to provide a genome-wide characterization of CTCF-binding sites and differentially CTCF-binding sites (DCBS) in rumen tissue during the weaning transition of four Holstein calves to uncover regulatory elements in rumen epithelial tissue using ChIP-seq. Our study generated 67,280 CTCF peaks for the before weaning (BW) and 39,891 for after weaning (AW). Then, 7401 DCBS were identified for the AW vs. BW comparison representing 0.15% of the cattle genome, comprising ~54% of induced DCBS and ~46% of repressed DCBS. Most of the induced and repressed DCBS were in distal intergenic regions, showing a potential role as insulators. Gene ontology enrichment revealed many shared GO terms for the induced and the repressed DCBS, mainly related to cellular migration, proliferation, growth, differentiation, cellular adhesion, digestive tract morphogenesis, and response to TGFβ. In addition, shared KEGG pathways were obtained for adherens junction and focal adhesion. Interestingly, other relevant KEGG pathways were observed for the induced DCBS like gastric acid secretion, salivary secretion, bacterial invasion of epithelial cells, apelin signaling, and mucin-type O-glycan biosynthesis. IPA analysis further revealed pathways with potential roles in rumen development during weaning, including TGFβ, Integrin-linked kinase, and Integrin signaling. When DCBS were further integrated with RNA-seq data, 36 putative target genes were identified for the repressed DCBS, including *KRT84*, *COL9A2*, *MATN3*, *TSPAN1*, and *AJM1.* This study successfully identified DCBS in cattle rumen tissue after weaning on a genome-wide scale and revealed several candidate target genes that may have a role in rumen development, such as *TGFβ*, *integrins*, *keratins*, and *SMADs*. The information generated in this preliminary study provides new insights into bovine genome regulation and chromatin landscape.

## 1. Introduction

The weaning period in calves is the transition from a milk-based diet to a solid feed diet, and dairy calves are traditionally weaned at the age of 1–3 months [1]. During this transition, the rumen is characterized by important structural changes, including increased capacity and surface area [2,3]. Appropriate rumen development is crucial for the dairy industry because it directly affects the calf’s nutrition and health, including feed intake and nutrient digestion [1]. However, few studies have been conducted to investigate genetic mechanisms affected by weaning transition in dairy calves [4,5,6,7,8].

*Cis*-regulatory elements (CREs), including promoters, enhancers, silencers, and insulators, play a critical role in gene expression. However, the accurate identification of these functional elements remains a challenge. The Encyclopedia of DNA Elements (ENCODE) project started in 2003 with the main goal of annotating the functional elements in the human genome [9], and later it was extended to other species [10,11,12]. Recently, efforts have been made to uncover regulatory elements in cattle using chromatin immunoprecipitation followed by sequencing (ChIP-seq) and assay of transposase accessible chromatin sequencing (ATAC-seq) approaches [8,13,14,15,16,17,18,19]. Chromatin mapping, such as ChIP-seq, has emerged as a powerful tool for understanding genome regulation and can generate large amounts of data on histone modifications across different tissues and cell types [20].

CCCTC-binding factor (CTCF) is a well-known transcription factor that binds the genome using its 11 zinc fingers. It is ubiquitously expressed and highly conserved in eukaryotic species [21]. CTCF has various crucial roles, including insulation activity, regulation and stabilization of chromatin architecture, transcriptional regulation (activator and repressor), gene imprinting, inactivation of the X chromosome in mammals, and DNA double-strand breaks repair [21,22,23,24]. In vertebrates, insulators or chromatin boundary elements are short DNA sequences that can block enhancers from interacting with unrelated genes or act as a barrier to protect genes [25,26]. CTCF can bind to insulators and result in blocking enhancer activity and repressing gene transcription [21,25]. Because of that, the identification of potential insulators is of great importance as it can help in understanding the activity of CREs and gene transcription [27].

Both histone modifications and CTCF-binding sites are key regulators that control chromatin structure and represent important functional genomic regions that may impact phenotypes [28]. CTCF ChIP-seq can identify CTCF-binding sites across the genome to understand the chromatin landscape and represents the first step in studying the role of CTCF in insulator functions [27,28].

Although large numbers of regulatory elements have been identified in cattle recently, many more regulatory non-coding elements are expected to be identified in additional tissues and conditions. A few studies have used CTCF ChIP-seq to identify functional regions in animals, such as mouse [29,30,31], sheep [32,33], and cattle [34]. A recent study in lactating dairy cows identified millions of functional regions in the cattle genome using ChIP-seq data from six tissues (heart, kidney, liver, lung, mammary, and spleen), including CTCF-binding sites and differential binding sites between tissues [18,34]. In a recent report from our group [8], different epigenetic marks in the cattle genome were identified, including CTCF, resulting in the first in vivo global map of regulatory elements and chromatin states in rumen tissue during weaning. Although this study identified thousands of CTCF-binding sites across the cattle genome [8], a deeper characterization of the CTCF-binding sites is needed, such as a comparison of the CTCF-binding sites before and after weaning and the functional annotation of these differentially CTCF-binding sites (DCBS) in pre-and post-weaning calves. Importantly, the identification of DCBS in the cattle genome is crucial to uncover functional and regulatory regions, particularly in rumen tissue, due to its key roles in cattle nutrition, health, and performance.

This study provides a genome-wide characterization of CTCF-binding sites in rumen tissue during the weaning transition in Holsteins. The main objectives of this study were to identify CTCF-binding sites and DCBS in pre-and post-weaning calves using the ChIP-seq approach and bioinformatics tools to uncover regulatory elements in rumen epithelial tissue. The characterization of the CTCF sites on a genome-wide scale provided here can be used as a resource for regulatory elements studies in cattle.

## 2. Results

### 2.1. Data Quality and CTCF Peaks

Two CTCF ChIP-seq datasets were generated (BW and AW). A total of 42,921,125 single-end reads were initially generated for BW, and 44,702,240 reads for AW (Table 1). Approximately 95% of the reads were aligned to the ARS-UCD1.2 cattle reference genome assembly [35], with an average of 41,695,284 reads mapped. On average, 0.13% of the reads were mapped to the mitochondrial genome; 13.02% were duplicated, and 14.37% had a mapping quality score (MAPQ) < 10. A total of 27,814,748 clean single-end reads were produced for BW, and 31,627,368 for AW, respectively.

The CTCF peaks were identified in the individual samples by the MACS2 [36]. A total of 59,442,116 clean reads were used for peak calling, generating 107,171 peaks (FDR < 0.05) for all samples with an average of 53,586 peaks and an average peak length of 446 (Table 2). There were more peaks identified in the BW (67,280, representing 1.06% of the cattle genome) than in the AW (39,891, representing 0.80% of the cattle genome). In addition, the quality of the CTCF ChIP-seq assay was assessed by calculating the fraction of reads in peaks (FRiP) for each sample. The BW FRiP was 0.22, and AW was 0.14. In addition, the specific number of peaks in each condition was obtained, showing a total of 33,333 BW-specific CTCF peaks and 5821 AW-specific peaks, and the consensus number of shared peaks was 34,459. Furthermore, the quality of the CTCF peaks was assessed, and the heatmap profiles of peaks relative to transcription start site (TSS) were generated (Appendix A).

### 2.2. Differentially CTCF-Binding Sites

The DCBS were identified using Diffreps [37]. A total of 26,871 DCBS (*p* < 0.05) was initially obtained for the AW vs. BW comparison (Table 3). Then, the DCBS were filtered based on FDR < 0.01 and |log_2_FC| > 1, and approximately 31% were retained. The 8249 DCBS were then mapped against a list of 72,710 merged CTCF peaks (Appendix A). Most of the DCBS were mapped in the merged peaks, totaling 7401 DCBS representing 0.15% of the cattle genome (Appendix A), including ~54% of induced sites (log_2_FC ≥ 1) and ~46% of repressed sites (log_2_FC ≤ −1) (Table 3 and Appendix A).

The annotation of the AW-induced and -repressed DCBS is presented in Figure 1 and Appendix A. Most of the induced and repressed DCBS were in distal intergenic regions (~61%), followed by introns (15% for the induced and 17% for the repressed) and promoters (9% for the induced and 19% for the repressed) (Figure 1A). In addition, approximately 8.7% of the induced DCBS were located on exons. Furthermore, the distribution of the induced and repressed sites relative to TSS was obtained (Figure 1B). The majority of the induced DCBS fall in 10–100 kb regions around the TSS, and most of the repressed sites fall in 10–100 kb and >100 kb regions around the TSS.

### 2.3. Functional Annotation of Differentially CTCF-Binding Sites

Gene Ontology (GO) enrichment and KEGG pathway analysis revealed significantly enriched (FDR < 0.05) GO terms and pathways (Appendix A). Both the induced and repressed DCBS were enriched for 366 GO biological process (BP) terms (e.g., regulation of cell migration and cell adhesion, response to TGFβ, epithelium development, and digestive tract morphogenesis) (Appendix A). The induced sites revealed specific important BP GO terms (e.g., regulation of cell growth, SMAD protein signal transduction, keratinocyte differentiation, and Integrin-mediated signaling pathway) (Appendix A). The repressed sites revealed specific BP GO terms related to the regulation of epithelial cell proliferation, TGFβ signaling pathway, digestive tract development, focal adhesion, digestive system development, cell/cell junction, and others (Appendix A). Figure 2 shows the top 30 BP GO terms for the DCBS, and Figure 3 shows the top 30 CC GO terms.

KEGG enrichment analysis revealed different pathways for the induced and repressed DCBS (Appendix A). Some of the important signaling pathways identified only for the induced sites were gastric acid secretion (Figure 4), apelin, VEGF, mucin-type O-glycan biosynthesis, MTOR, Rap1, growth hormone, bacterial invasion of epithelial cells, and salivary secretion (Appendix A). Relevant pathways identified only for the repressed sites were TGFβ (Figure 5), cell cycle, PPAR, ECM-receptor interaction, and butanoate metabolism (Appendix A). Furthermore, 20 common pathways were identified, including adherens junction (Appendix A), focal adhesion (Appendix A), EGFR tyrosine kinase inhibitor resistance, WNT, and sphingolipid (Appendix A).

### 2.4. IPA Results

IPA was used to obtain pathways from the DCBS. Significant networks (network score > 20) from the induced sites were mainly related to cellular assembly and organization, morphology, and maintenance, and networks from the repressed sites were mainly related to the cellular cycle, signaling, movement, assembly, and organization (Appendix A). In addition, several significant canonical signaling pathways (*p* < 0.01) were identified for the induced DCBS (e.g., TGFβ, ILK, Integrin, VEGF, apelin muscle, cholecystokinin/gastrin-mediated), and for the repressed sites (e.g., TGFβ, ILK, epithelial adherens junction, ERK/MAPK, WNT/β-catenin, HIPPO) (Appendix A). Significant upstream regulators (*p* < 0.01) such as growth factors, kinases, TFs, and transmembrane receptors were identified for the induced DCBS (e.g., TGFβ1, VEGFA, TGFBR2, MTOR, SMAD2-5, SMARCA4/5, ITGB1, ITGA5), and for the repressed sites (e.g., TGFβ1/3, VEGFA, TGFBR1/2, MTOR, SMAD2-4/7, SMARCA4, ITGB1). More details can be found in Appendix A.

### 2.5. Identification of Putative Transcription Factor Binding Sites

HOMER [38] was used to identify enriched motifs (*p*-value ≤ 0.01) and putative transcription factor binding sites (TFBS) in the induced and repressed DCBS. A total of 28 and 91 enriched motifs were identified for the induced and repressed DCBS, respectively (Appendix A). The top ten identified TFBS, ranked according to *p*-values, for the induced sites were CTCF, BORIS, ZEB2, ZEB1, TAL1, E2A, NEUROD1, TBX21, MITF, and TBX5, and for the repressed sites were CTCF, BORIS, BCL11A, TGIF2, TBX5, FLI1, ELK1, ETV1, ZIC3, and GABPA. In addition, when the results were ranked according to the percentage of target sequences with motifs, for the induced sites, the top ten were THRB, TBX5, HEB, ZEB1, E2A, ASCL1, ZNF416, EOMES, ASCL2, and ZEB2, and for the repressed sites were SCL, PITX1, TGIF2, TBX5, BORIS, CTCF, PTF1A, HIC1, SMAD3, and TGIF1.

### 2.6. RNA-Seq Integration with CTCF ChIP-Seq Data

To identify putative target genes, DCBS were integrated with previous RNA-seq data [8] with the BETA tool [39]. BETA integrates ChIP-seq data with differential gene expression data to infer direct target genes and to predict whether the factor has activating or repressive function [39]. The analyses were done for the induced and repressed DCBS after weaning separately and for the combined induced + repressed DCBS. No significant results were identified for the separate analyses (Appendix A). The functional impact of combined DCBS was gene-repressing (*p* = 0.000606, Appendix A), and 36 target genes (rank value/*p* < 0.01) were identified (Appendix A). Interesting down-regulated target genes found were *Keratin 84* (*KRT84*), *Collagen type IX alpha 2 chain* (*COL9A2*), *Matrilin 3* (*MATN3*), *Tetraspanin 1* (*TSPAN1),* and *Apical junction component 1 homolog* (*AJM1*).

### 2.7. Visualization of CTCF Signals in Selected Genes

The CTCF peaks from BW and AW conditions and the induced and repressed DCBS after weaning were examined in the IGV tool [40] for selected genes related to potential roles in rumen development during weaning in calves, such as cellular adhesion and cell proliferation. The selected genes were four *Integrin Subunit Beta* genes (*ITGBs*), including *ITGB1* (Appendix A), *ITGB2* (Appendix A), *ITGB4* (Appendix A), and *ITGB5* (Appendix A). In addition, nine *keratin* genes were selected, including *KRT9* and *KRT14* (Appendix A), *KRT32, 35,* and *36* (Appendix A), *KRT80* (Appendix A), and *KRT83*, *KRT85,* and *KRT89* genes (Appendix A).

## 3. Discussion

CTCF is an essential protein that binds with tens of thousands of genomic sites and can form chromatin loops and defines the boundaries between active and inactive DNA [41], and has essential roles including insulator, chromatin remodeler, transcriptional regulation, genome organization, and others [21,22]. CTCF also interacts with cohesins, and both regulate the chromatin loop stability [42]. CTCF identification in the cattle genome is crucial to help to unravel regulatory regions, especially in rumen tissue, due to its essential roles in cattle nutrition and health. Because of that, the objective of this study was to identify and characterize CTCF-binding sites during the weaning transition using the ChIP-seq approach and bioinformatics tools to uncover regulatory elements in rumen epithelial tissue.

### 3.1. Identification of CTCF Peaks

The ENCODE guidelines for TF ChIP-seq projects recommend at least 20 million usable fragments for each sample in human and mouse experiments (https://www.encodeproject.org/chip-seq/transcription-factor-encode4/ (accessed on 12 August 2022)). Our study generated >27 million clean reads for each condition that includes two samples per pool, and the quality per sample was above the recommended threshold. However, at least two replicates per sample are recommended for this type of study according to ENCODE. Because of that, one major limitation of this study was the sample size with only four samples. In addition, ENCODE recommends the fraction of reads in peaks (FRiP score) as an additional metric to verify the quality of the TF ChIP-seq experiment, and our results of FriP score >0.1 for the two conditions indicate an acceptable quality for the number of reads in peaks [43].

This study generated a total of 67,280 CTCF peaks for the BW condition, representing 1.06% of the cattle genome, and 39,891 peaks for the AW, representing 0.80% of the cattle genome (bosTau9). These results are in accordance with a previous study in bovine rumen tissue from our group, where the CTFC peaks identified for BW and AW represented approximately 0.6% of the cattle genome (bosTau8) [8]. Another study in cattle identified CTCF peaks in the liver, lung, mammary gland, kidney, heart, and spleen in lactating Holstein cows, totaling an average of 456,881 CTCF peaks for all samples and tissues (with a minimum number of ~60,000 peaks and a maximum number of 790,000 peaks per sample), representing ~7% of the cattle genome (bosTau8), showing a higher number of peaks compared to our study probably due to the higher number of samples and number of reads [34]. However, their samples presented a great variability in the number of CTCF peaks depending on the tissue and replicate [34]. Furthermore, a study on sheep identified CTCF-binding sites that represented 0.7% of the sheep genome [32].

This study demonstrated that ChIP-seq was able to identify thousands of CTCF-binding sites. However, this study is a preliminary trial due to the lack of replicates and small sample size, and future studies with larger sample sizes and replicates are needed to confirm these findings.

### 3.2. Differentially CTCF-Binding Sites

In addition, to identify CTCF peaks across the cattle genome, our goal was also to detect DCBS by comparing AW vs. BW conditions to detect the regions that have significant enrichment in the CTCF-binding sites AW. The Diffreps [37] was utilized to scan the genome for enrichment regions and detect DCBS. Additional filtration steps were performed to ensure the quality of the DCBS, including the removal of the regions that did not overlap with MACS2 peaks. Similar quality control of the DCBS was done in mice [29]. After these filtration steps, we were able to identify 3812 induced DCBS and 3229 repressed DCBS after weaning. We utilized these DCBS for the downstream analysis to identify differences in the biological functions of the induced and repressed sites.

The annotation of the induced and repressed DCBS AW revealed different patterns. For example, more repressed sites were in promoter regions compared to the induced sites (+10%), while more induced sites were in exons (+7%), UTR (+2.7%), and downstream (+2.2%). However, most of the induced and repressed DCBS are in distal intergenic regions (~60%), in accordance with previous results in sheep [33] and humans [44], showing a potential role as insulators, indicating that these regions may also act as insulators by blocking enhancers and silencers present nearby these sequences [27]. In addition, a significant percentage of the sites are located within genes (~24% for the induced and 19% for the repressed sites). In addition, the distribution of the induced and repressed DCBS relative to TSS also showed differences, resulting in a higher percentage of the repressed DCBS in 0–1 kb and >100 kb. Most of the induced DCBS fall in 10–100 kb. The induced and repressed DCBS were detected on both sides of the TSS. These results indicated that the majority of the DCBS are located far from the TSS, also in accordance with a previous study in vertebrates [27]. We demonstrate here that the CTCF-binding site localization is consistent with its potential role as an insulator. Interestingly, the repressed sites after weaning appeared to have higher proximity to its TSS, suggesting these decreased CTCF binding may mediate these genes’ expression.

### 3.3. Enrichment and Pathway Results of DCBS

Gene ontology enrichment results revealed that there are many common GO terms from the induced and the repressed DCBS, mainly related to cellular migration, proliferation, growth, and differentiation; cellular adhesion and junction; tissue development; digestive tract morphogenesis, and response to TGFβ and growth factor. Weaning transition results in several anatomical changes in the rumen of calves [2,3]. This study shows that CTCF-binding sites may have important effects on the regulation of cellular and tissue growth, and cellular adhesion. In addition, our study revealed that induced and repressed DCBS presented different GO terms, indicating that there is a significant shift of the CTCF binding during weaning. Previous studies in cattle revealed similar biological processes as a result of the weaning, such as cell adhesion and cell migration, when studying chromatin accessibility [19], or the landscape of chromatin states [8].

Adherens junction and focal adhesion were common KEGG pathways for the induced and repressed DCBS, as previously observed in the GO enrichment analysis mentioned above. Interestingly we observed relevant biological KEGG pathways for the induced sites related to the rumen and gastrointestinal development, such as gastric acid secretion, salivary secretion, bacterial invasion of epithelial cells, apelin signaling pathway, and mucin-type O-glycan biosynthesis.

In cattle, gastric acid is released by the abomasum, helping the hydrolysis of microbial and dietary protein. Saliva in cattle is essential for food ingestion, nutrient release, and circulation, and is an important pH buffer in the rumen, especially in high-concentrate feeding [45,46]. The weaning period in calves results in changes in the diet, and these modifications may affect salivary [46] and gastric acid secretion [47]. Mucin-type O-glycans (O-glycans) are proteins expressed on mucosal surfaces, and they are important components of the gastrointestinal mucus and protect epithelial cells against damage or infections [48]. There are few studies evaluating mucins in cattle. A previous study in cattle identified 15 mucin-encoding genes in the cattle genome and most of them were transcribed in the bovine gastrointestinal tract, including rumen [49]. As previously shown in humans, mice, rats, and pigs, mucin levels increase between birth and weaning, and decrease after weaning [50]. However, more studies are necessary to understand the mucin levels before and after weaning in cattle and their effects on the gastrointestinal tract. Furthermore, apelin has been related to several biological functions, including glucose, lipid and water metabolism, embryonic development, and control of food intake [51]. In addition, it has been reported that apelin has receptors in the gastrointestinal tract, and several apelin peptides are present in the bovine colostrum and might be involved in gastrointestinal tract development [52,53].

We also detected biological KEGG pathways present only for the repressed DCBS, such as the TGFβ signaling pathway and butanoate (or butyrate) metabolism. A similar result was found before in our recent study evaluating open chromatin regions from the same samples during weaning in calves [19], where repressed differentially accessible regions after weaning were identified on the *TGFβ2* and *TGFBR2* genes, suggesting that the TGFβ pathway may be more active before and during weaning to promote the rumen development. Butyrate is a short-chain fatty acid produced by ruminal fermentation involved in rumen development and cell proliferation in cattle, and butyrate supplementation in preweaning calves promotes ruminal epithelium maturation [54].

IPA analysis also revealed similar pathways of biological importance, including TGFβ signaling, Integrin-linked kinase (ILK) signaling, and Integrin signaling. Integrins are transmembrane receptors related to cell adhesion and signal transduction [55], and in cattle, the *ITGB1* gene was related to rumen epidermal proliferation response [56]. Integrin pathways were also identified previously in cattle related to open chromatin regions during weaning [19].

### 3.4. Putative Transcription Factor Binding Sites

We performed motif enrichment analysis to further identify TFBS in the DCBS. CTCF, BORIS, E2A, MITF, ASCL2, and NR5A2 were some of the relevant TFBS for the induced sites. BORIS (CTCFL) is the paralog of CTCF and might have a role in unfolding the chromatin preceding transcription [57]. E2A is involved in cell growth, commitment, and differentiation, and also in the mesenchymal-epithelial transition [58]. MITF has several roles, including cell differentiation, survival and proliferation, cell metabolism, and DNA damage repair [59]. ASCL2 is a major regulator of intestinal stem cells [60]. NR5A2 has a role in cellular proliferation, bile acid homeostasis, and steroidogenesis [61]. CTCF and BORIS were also identified for the repressed sites, and TGIF1/2, ETV1, SMAD2/3, ELK1, AP1, ATF3, ETS, and others. TGIF1 and TGIF2 are transcriptional repressors. For example, TGIF1 suppresses the TGF-β signaling by SMAD2-SMAD4 interaction [62]. SMAD2 and 3 are mediators of the TGF-β signaling and affect cell growth, migration, proliferation, and cell death [63]. ELK1 is involved in gene expression control and regulation of the basal transcription machinery [64]. AP1, ATF3, and ETS are involved in cell proliferation and differentiation [65,66]. ETV1 is a member of the ETS family and plays a role in cell proliferation, migration, and differentiation [67]. Five TFs previously identified in cattle rumen tissue during weaning [4] were also identified in our study from the repressed sites—ATF3, ETS1, FOS, JUNB, and KLF4.

### 3.5. RNA-Seq Integration with CTCF-Binding Sites

CTCF-binding sites were further integrated with RNA-seq data [8], and 36 target genes for the repressed sites were identified, including *KRT84, COL9A2, MATN3, TSPAN1*, and *AJM1.* Keratins are present in epithelial cells, including the rumen surface, resulting in protection from physical damage of ingested plant material [68]. Rumen mucosa keratinization levels increase when calves are fed with a solid diet, especially a high-concentrate or high-forage diet, resulting in keratinization of rumen papillae and thickening of the *Stratum corneum* [69,70,71]. Collagen fibrils are present in the rumen epithelium mucosa and the core of the rumen papillae [72]. However, changes in the rumen collagen content during weaning have not been reported. Combining the literature information with our findings, we can hypothesize that keratin and collagen gene expression levels in the bovine rumen epithelium may be affected by weaning and diet changes. MATN3 is part of the matrilins protein family that is highly expressed in cartilage and other extracellular matrices and binds to different types of collagen fibrils [73]. In a study in Hanwoo and Holstein cattle, a copy number variation (CNV) in the *MATN3* gene was mainly deleted in Holstein [74]. The authors concluded that this gene is related to osteoarthritis, which is a risk factor in dairy cattle. TSPAN1 is involved in cell proliferation, invasion, and survival [75]. The *AJM1* gene is predicted to be involved in cell/cell junction organization [76].

### 3.6. CTCF Signals in Selected Genes

In addition, we selected genomic regions for visualization of genes related to potential functions in rumen development during weaning, such as cellular adhesion and cell proliferation. Four regions showing CTCF peaks and induced and repressed DCBS were selected for four *ITGB* genes, including *ITGB1, ITGB2*, *ITGB4*, and *ITGB5* on chromosomes 1, 13, and 19. Seven induced regions and one repressed region were observed in our study for the *Integrin beta* genes. Integrins have roles in cell adhesion, differentiation, and migration [55,77,78]. In a study in cattle, a repressed differentially accessible region after weaning was identified in the *ITGB1* gene [19]. We also identified a repressed DCBS after weaning in the *ITGB1* gene, showing that this gene was induced BW. In addition, four regions were detected from nine keratin genes on chromosomes 5 and 19—*KRT9*, *KRT14*, *KRT32*, *KRT35*, *KRT36*, *KRT80*, *KRT83*, *KRT85*, and *KRT89* genes. Twelve DCBS were observed, including ten induced sites and two repressed sites. As mentioned before, keratins are found on the rumen surface [68], and keratinization of rumen papillae can increase depending on the diet type [69,70,71]. Weaning in calves represents a change in diet, from a milk-based to a solid diet, and this feed change will probably affect the keratinization of rumen papillae as well.

## 4. Materials and Methods

### 4.1. Rumen Epithelial Tissue Collection

The Beltsville Area Animal Care approved animal care and tissue isolation work (Committee Protocol Number: 07-025). Four Holstein bull calves were utilized: two (before weaning, BW) two-week-old calves were fed with milk replacer only (MRO-Cornerstone 22:20, Purina Mills, St. Louis, MO, USA; 22.0% crude protein, 20.0% crude fat, 0.15% crude fiber, 0.75 to 1.25% Ca, 0.70% P, 66,000 IU/kg vitamin A, 11,000 IU/kg vitamin D3, and 220 IU/kg vitamin E) for two weeks, while the other two eight-week-old calves (six-week with MRO only and two weeks with MRO and orchardgrass hay (after weaning (AW)) were fed with MRO for six weeks, followed by a combination of milk replacer and grain-based commercial calf starter for four weeks. Calves were euthanized by captive bolt followed by exsanguination at Day 14 or Day 70 to represent development at two stages of weaning on a grain concentrate diet. Rumen epithelial tissue was collected from the anterior portion of the ventral sac of the rumen beneath the reticulum and below the rumen fluid layer at slaughter. The epithelial layer of the rumen tissue was separated manually from the muscular layer. After being rinsed in tap water to remove residual feed particles, samples were further rinsed in ice-cold saline and snap-frozen in liquid nitrogen before moving to −80 °C for future use. Then, samples were fixed in RNAlater RNA stabilization solution (Life Technologies, Grand Island, NY, USA) according to the manufacturer’s instructions and stored at −80 °C. More details can be found in our previous publication [8]. The samples from two calves for each condition were polled for the subsequent sequencing–

### 4.2. CTCF ChIP-Sequencing

CTCF ChIP-seq from the two pooled samples (two from BW and two from AW) was performed by Active Motif, Inc. (Carlsbad, CA, USA). The DNA integrity was verified using the Agilent Bioanalyzer 2100 (Agilent; Palo Alto, CA, USA). The DNA was then processed, including end repair, adaptor ligation, and size selection, using an Illumina sample prep kit following the manufacturer’s instructions (Illumina, Inc., San Diego, CA, USA). The DNA libraries were then sequenced (75 bp) on an Illumina HiSeq 2500 platform (Illumina, San Diego, CA, USA).

### 4.3. ChIP-Seq Data Processing and Mapping

First, sequence reads were examined for quality using FastQC v.0.11.9 (https://www.bioinformatics.babraham.ac.uk/projects/fastqc/ (accessed on 12 August 2022)). Reads were then aligned to the ARS-UCD1.2 cattle reference genome assembly [35] using BWA v.0.7.17 [79]. Unmapped reads, reads mapped to multiple locations, reads with MAPQ < 10, and reads located on the mitochondrial chromosome were removed by SAMtools v.1.9 [80]. Duplicate reads were removed with Picard v.2.22.3 (https://broadinstitute.github.io/picard/ (accessed on 12 August 2022)).

### 4.4. ChIP-Seq Peak Calling and Quality Check

Individual peaks were called by MACS2 v.2.2.7.1 [36] using default parameters and FDR < 0.05. Peaks located on chromosome X or unplaced were removed. The fraction of all mapped reads in enriched peaks (FRiP) was obtained for each sample. BEDTools v.2.25.0 [81] with intersect function was used to obtain the specific number of peaks for each condition.

### 4.5. Identification of Differentially CTCF-Binding Sites

DiffReps v.1.55.6 [37] was used to identify the DCBS between AW vs. BW. A defined window of 200 bp and a G-test (*p* < 0.05) were used. The DCBS were filtered with an FDR < 0.01 and |log_2_FC| >1. Then, the filtered regions were compared against the identified MACS2 peaks. To do this integration, first, the identified peaks were merged into a single list of non-overlapping peaks by BEDtools v.2.25.0 [81] with the intersect function. The DCBS that coincided with MACS2 peaks in at least one sample were further analyzed. The induced and repressed DCBS were separated based on their log_2_ fold change values and used for the downstream analyses.

### 4.6. DCBS Annotation

The induced and repressed DCBS were annotated with the *annotatePeak* function from the ChIPseeker package [82]. Promoter regions were defined as ±2 kb from the TSS. In addition, the *plotDistToTSS* function from the ChIPseeker [82] was used to plot the distance of the DCBS around the TSS.

### 4.7. Enrichment and Pathway Analysis

Genes annotated from the induced, and repressed DCBS were searched using ShinyGO v.0.76 [83] to obtain enriched biological process (BP), cellular component (CC), and molecular function (MF) terms, and KEGG pathways using FDR < 0.05. QIAGEN Ingenuity Pathway Analysis (IPA) v.68752261 [84] was used with default parameters to find signaling and metabolic pathways, including canonical pathways (*p* < 0.01), upstream regulators (*p*-value of overlap < 0.01), and molecular networks (network score > 20).

### 4.8. Motif Enrichment

The enriched motifs were identified with HOMER v.4.11 [38] from the induced and repressed DCBS with *findMotifsGenome* function (*p*-value ≤ 0.01 and >5% of target sequences with motif).

### 4.9. RNA-Seq Integration with CTCF ChIP-Seq Data

To investigate gene expression and regulatory networks and compare with DCBS, previously described RNA-seq data from weaning (six samples with three biological replicates) were utilized [8] to obtain differentially expressed genes (DEG) (data is available at the NCBI SRA database, BioProject ID: PRJNA658627). Four samples from the RNA-seq data correspond to the same samples utilized in this study. RNA-seq clean reads (Q > 20) were aligned to the ARS-UCD1.2 cattle genome assembly [35] with STAR v.2.7 [85], and gene expressions and DEG were obtained using Cufflinks v.2.2.1 [86]. The integration of DEG and DCBS was performed with the BETA tool v.1.0.7 [39] using its basic function and *p*-value of 0.01. The analyses were performed for the induced and repressed DCBS separately and jointly with the combined induced + repressed DCBS. The *p*-values from BETA (rank product) are estimated by the Kolmogorov-Smirnov test comparing the regulatory potential of up-regulated, down-regulated, and background genes. Genes with *p*-values/rank product < 0.01 were considered significant target genes.

### 4.10. Visualization of the CTCF Signals in Selected Genes

The bigwig files from BW and AW were generated from MACS2 bedGraph files using the *bedGraphToBigWig* tool [87]. Then, the bigwig files and bed files from the induced and repressed DCBS were directly visualized using the Integrative Genomics Viewer (IGV) [40] for selected genes of interest related to potential roles in rumen development during weaning, such as cell proliferation and cellular adhesion. The genes selected were *integrins* and *keratins*.

## 5. Conclusions

Although several regulatory elements have been identified in cattle, a deeper characterization of the CTCF-binding sites is needed. The identification of CTCF-binding sites in rumen epithelial tissue is crucial for uncovering regulatory regions, due to its major importance for cattle nutrition and health. This study successfully identified thousands of CTCF-binding sites and differentially CTCF-binding sites in the cattle genome in rumen tissue of Holstein pre-and post-weaning calves. However, one major limitation of this study was the sample size of only four samples. Future studies with larger sample sizes are needed to confirm these findings. Functional analyses were conducted for the differentially CTCF-binding sites, including gene ontology enrichment, pathways, motif enrichment, and integration with RNA-seq to identify transcription factors and putative candidate target genes for rumen development during weaning. This deep characterization of CTCF-binding sites revealed several candidate target genes that may have a role in rumen development, such as *TGFβ*, integrins, keratins, and *SMADs*, resulting in a valuable resource of regulatory elements in cattle with insights into the chromatin landscape.

## Figures and Tables

**Figure 1 ijms-23-09070-f001:**
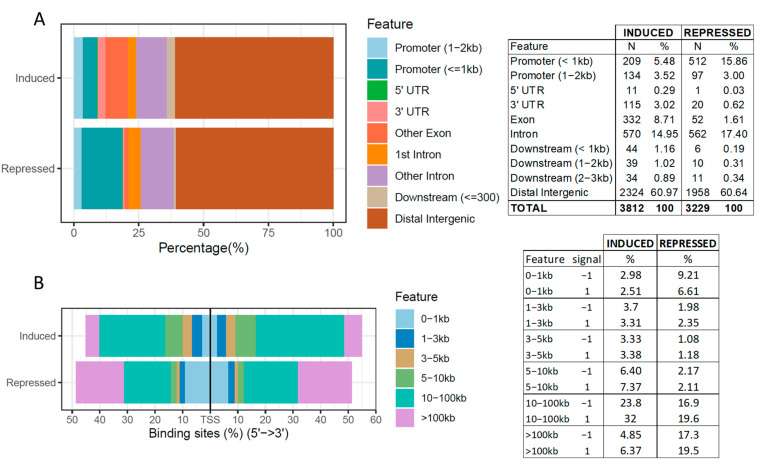
(**A**) Annotation of induced and repressed differentially CTCF-binding sites (DCBS) after weaning. (**B**) Distribution of the induced and repressed DCBS after weaning relative to TSS.

**Figure 2 ijms-23-09070-f002:**
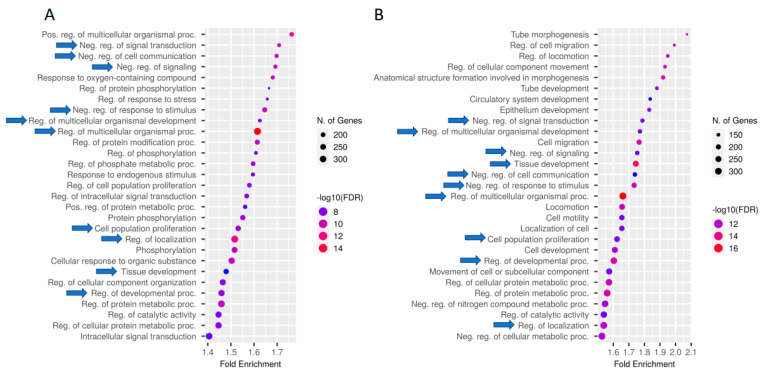
GO enrichment results for the induced (**A**) and repressed (**B**) differentially CTCF-binding sites after weaning. The top 30 hits of GO terms are shown in this figure. Biological processes are ranked according to the fold enrichment values. Bubble colors represent the *p*-value of the False Discovery Rate (FDR). The most significant processes are highlighted in red, and the less significant processes are highlighted in blue according to log_10_ (FDR) values. Bubble sizes indicate the number of genes. Blue arrows indicate GO terms present in both induced and repressed sites.

**Figure 3 ijms-23-09070-f003:**
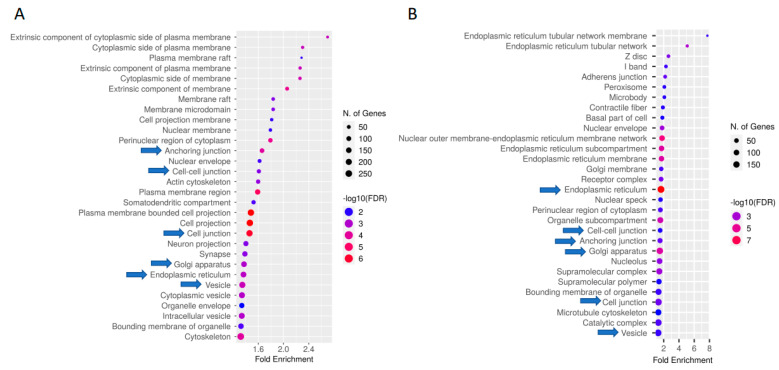
GO enrichment analysis for the induced (**A**) and repressed (**B**) differentially CTCF-binding sites after weaning. The top 30 hits of GO terms are shown in this figure. Cellular components are ranked according to the fold enrichment values. Bubble colors represent the *p*-value of the false discovery rate (FDR). The most significant processes are highlighted in red, and the less significant processes are highlighted in blue according to log10 (FDR) values. Bubble sizes indicate the number of genes. Blue arrows indicate GO terms present in both induced and repressed sites.

**Figure 4 ijms-23-09070-f004:**
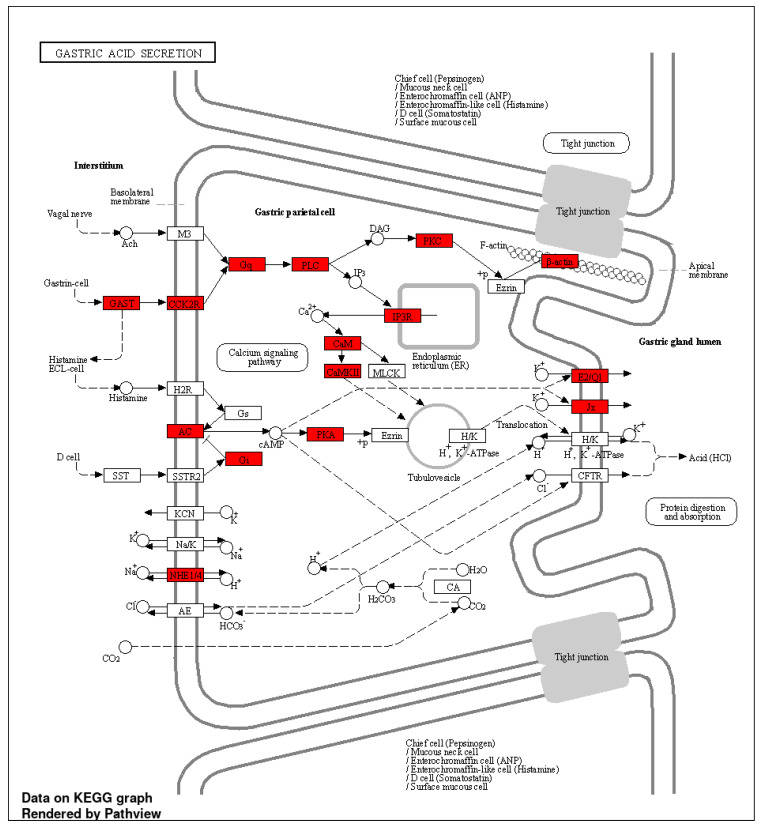
KEGG gastric acid secretion pathway obtained from the induced differentially CTCF-binding sites (DCBS). Genes from the induced DCBS are highlighted in red.

**Figure 5 ijms-23-09070-f005:**
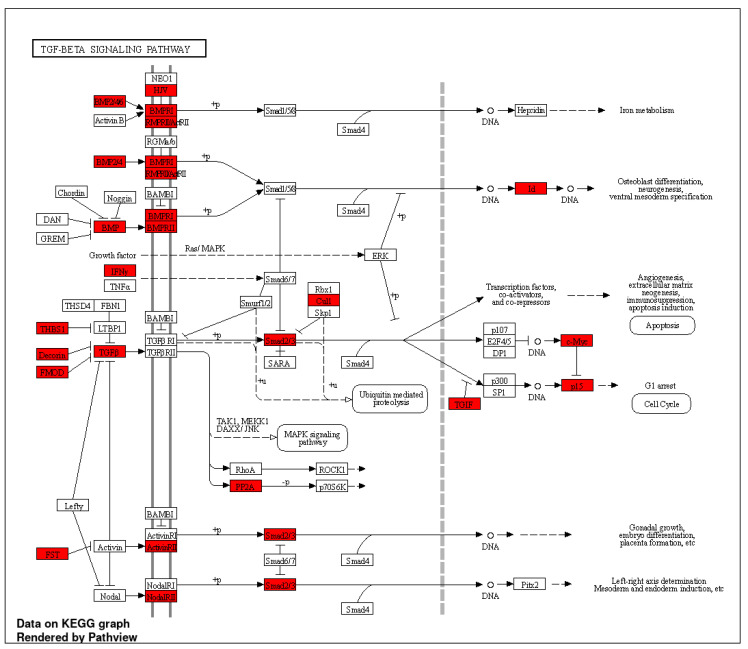
KEGG TGF-β signaling pathway obtained from the repressed differentially CTCF-binding sites (DCBS). Genes from the repressed DCBS are highlighted in red.

**Table 1 ijms-23-09070-t001:** CTCF ChIP sequence read statistics showing the number of reads, number and percentages of reads mapped, mitochondrial reads, duplicate reads and reads with mapping quality < 10, and the number of clean reads used for peak calling.

Condition	N of Reads	Mapped Reads (bosTau9)	Mapped Reads%	MT Reads	MT Reads% ^1^	Duplicate Reads	Duplicate Reads% ^1^	MAPQ < 10 Reads	MAPQ < 10 Reads% ^1^	Clean Reads ^2^
BW	42,921,125	41,327,290	96.29	38,772	0.09	6,695,091	16.20	6,316,659	15.28	27,814,748
AW	44,702,240	42,063,278	94.10	72,080	0.16	4,140,259	9.84	5,655,767	13.45	31,627,368
Total	87,623,365	83,390,568	-	110,852	-	10,835,350	-	11,972,426	-	59,442,116
Average	43,811,683	41,695,284	95.19	55,426	0.13	5,417,675	13.02	5,986,213	14.37	29,721,058

BW: before weaning. AW: after weaning. MT: mitochondrion. ^1^ Percentages were calculated considering the total number of mapped reads. ^2^ Reads uniquely mapped, with MAPQ > 10, no duplicate reads or reads located on MT chromosome.

**Table 2 ijms-23-09070-t002:** Peak calling metrics showing the total number of clean reads used to call peaks and calculate the fraction of reads in peaks (FRiP), number of CTCF peaks (FDR < 0.05), number of assigned reads in peaks, FRiP, an average of peak lengths, and the proportion of peaks near TSS (±3 kb, %).

Condition	Clean Reads ^1^	Clean Reads Used for FRiP ^2^	CTCF Peaks ^2^	N of Assigned Reads in Peaks ^2^	FRiP ^3^	Average Peak Length	Peaks Near TSS (±3 kb, %)
BW	27,814,748	27,066,621	67,280	6,193,989	0.22	393	17.56
AW	31,627,368	30,864,883	39,891	4,401,267	0.14	498	20.61
Total	59,442,116	57,931,504	107,171	10,595,256	-	-	-
Average	29,721,058	28,965,752	53,586	5,297,628	0.18	446	19.09

BW: before weaning. AW: after weaning. ^1^ Reads uniquely mapped, with MAPQ > 10, no duplicate reads or reads located on MT chromosome. ^2^ Reads located on chromosomes X and unplaced were not included. ^3^ Fraction of reads in peaks.

**Table 3 ijms-23-09070-t003:** Number of differentially CTCF-binding sites (DCBS) after weaning showing the different steps of filtration, including the number of induced and repressed sites.

Filtration Steps of the After-Weaning DCBS	N	%
Initially detected DCBS (*p* < 0.05)	26,871	100
Filtered DCBS (FDR < 0.01 and |log_2_FC| > 1)	8249	30.70
Filtered DCBS that overlapped with peaks	7041	26.20
Total of filtered DCBS	7041	100
Induced sites with log_2_FC ≥ 1	3812	54.14
Repressed sites with log_2_FC ≤ −1	3229	45.86

## Data Availability

All high-throughput sequencing data analyzed in this study are deposited in NCBI. RNA-seq data are publicly available in the NCBI SRA database (BioProject ID: PRJNA658627). All CTCF ChIP-seq data were submitted to NCBI, SRA database (BioProject ID: PRJNA672996).

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
