# Peer review of "Differentially CTCF-Binding Sites in Cattle Rumen Tissue during Weaning"

_ijms, 2022, doi:10.3390/ijms23169070_

Round 1

Reviewer 1 Report

It is a twin manuscript to the previously published titled „Characterization of Accessible Chromatin Regions in Cattle Rumen Epithelial Tissue during Weaning”.

The authors even used the same research material.

The article is interesting and seeks to clarify the involvement of the transcription factor CCCTC during weaning in bull calves.

However, some aspects demands carifiacation and/or/and additions before their publication. In details:

1.       The number of experimental animals – 4 bulls calves is not representative for statistical analyses and demands increasing.

2.       How old were the animals at the beginning of the experiment?

3.       What was the idea for choosing the genes for experiment 2.10? Please write the selected genes in the desription of the experiment.

4.       Discussion i too long, my suggestion is division it for paragraphs with subtitles.

Author Response

Reviewer 1

It is a twin manuscript to the previously published titled "Characterization of Accessible Chromatin Regions in Cattle Rumen Epithelial Tissue during Weaning". The authors even used the same research material.

The article is interesting and seeks to clarify the involvement of the transcription factor CCCTC during weaning in bull calves.

However, some aspects demands clarification and/or/and additions before their publication. In details:

  1. The number of experimental animals – 4 bulls calves is not representative for statistical analyses and demands increasing.

R: Thanks for your comment. We agree that four is a small sample size. However, we don't have additional samples to analyze currently, and even with small sample size, interesting findings were reported, and these findings can be used as preliminary results for future studies. Also, we conducted the recommended quality check of the four samples, and the results were in accordance with the ENCODE guidelines, showing that the quality per sample was above the recommended threshold. But we agree that the major limitation of this study is the small sample size, and to clarify that, we added more information highlighting this limitation in the Abstract, Discussion (Section 4.1), and Conclusions.

  1. How old were the animals at the beginning of the experiment?

R: Thanks for pointing that out. The animals were two-week-old at the beginning of the experiment, and this information was added in the Material and Methods section 2.1.

  1. What was the idea for choosing the genes for experiment 2.10? Please write the selected genes in the desription of the experiment.

R: Thanks for pointing that out. We selected a few genes for the CTCF signal visualization based on their potential roles in rumen development during weaning, such as cell proliferation and cellular adhesion. To clarify, we included the genes selected and additional information in the Material and Methods Section 2.10.

  1. Discussion is too long, my suggestion is division it for paragraphs with subtitles.

R: Thanks for pointing that out. We added the subtitles and improved the discussion by removing some redundant parts.

Reviewer 2 Report

a) The abstract is adequately written, but I suggest shortening it somewhat.

b) I propose to remove the "List of Abbreviations" from the manuscript (after 5. Conclusions). They are explained in the text of the manuscript.

Author Response

a) The abstract is adequately written, but I suggest shortening it somewhat.

R: Thanks for your suggestion. We shortened the abstract as suggested.

b) I propose to remove the "List of Abbreviations" from the manuscript (after 5. Conclusions). They are explained in the text of the manuscript.

R: Thanks for your suggestion. The list of abbreviations was removed.

Round 2

Reviewer 1 Report

The authors proved their manusript according to my suggestion. Even if the number of animals is still same - N=4, it can be published as a preliminary research.